# Endo-SemiS: Towards Robust Semi-Supervised Image Segmentation for Endoscopic Video

**Hao Li**[*][1]                                                                                     HAO.LI.1@VANDERBILT.EDU
**Daiwei Lu**[1]                                                                                    DAIWEI.LU@VANDERBILT.EDU
**Xing Yao**[1]                                                                                      XING.YAO@VANDERBILT.EDU
**Nicholas Kavoussi**[2]                                                          NICHOLAS.L.KAVOUSSI@VUMC.ORG
**Ipek Oguz**[1]                                                                                     IPEK.OGUZ@VANDERBILT.EDU
[1] *Vanderbilt University*
[2] *Vanderbilt University Medical Center*

**Editors:** Accepted for publication at MIDL 2026

## Abstract

In this paper, we present **Endo-SemiS**, a semi-supervised segmentation framework for providing reliable segmentation of endoscopic video frames with limited annotation. Endo-SemiS uses 4 strategies to improve performance by effectively utilizing all available data, particularly unlabeled data: (1) Cross-supervision between two individual networks that supervise each other; (2) Uncertainty-guided pseudo-labels from unlabeled data, which are generated by selecting high-confidence regions to improve their quality; (3) Joint pseudo-label supervision, which aggregates reliable pixels from the pseudo-labels of both networks to provide accurate supervision for unlabeled data; and (4) Mutual learning, where both networks learn from each other at the feature and image levels, reducing variance and guiding them toward a consistent solution. Additionally, a separate corrective network that utilizes spatiotemporal information from endoscopy video to improve segmentation performance. Endo-SemiS is evaluated on two clinical applications: kidney stone laser lithotomy from ureteroscopy and polyp screening from colonoscopy. Compared to state-of-the-art segmentation methods, Endo-SemiS substantially achieves superior results on both datasets with limited labeled data. The code is publicly available at `https://github.com/MedICL-VU/Endo-SemiS`

**Keywords:** Comprehensive supervision, uncertainty-guided pseudo-label, spatiotemporal

## 1. Introduction

Endoscopic image segmentation poses unique challenges, including large variations in image quality and appearance, which may be caused by motion blur, fluctuating lighting conditions (Li et al., 2025), and often fluid-filled environments (Setia et al., 2023), as well as domain shifts (Ali et al., 2023). These effects are illustrated in Fig. 1, which shows blur, bleeding, debris, occlusions, and cross-site or cross-device appearance changes in ureteroscopy and colonoscopy images. The limited availability of manual labels further complicates the task.

Semi-supervised learning (SSL) approaches provide a potential solution by effectively leveraging information from unlabeled data (Sohn et al., 2020; Chen et al., 2021; Luo et al., 2022a,b; Yang et al., 2023; Tarvainen and Valpola, 2017; Wang et al., 2024). These methods construct supervision signals for unlabeled samples from the predictions of the model

---

[*] Corresponding author

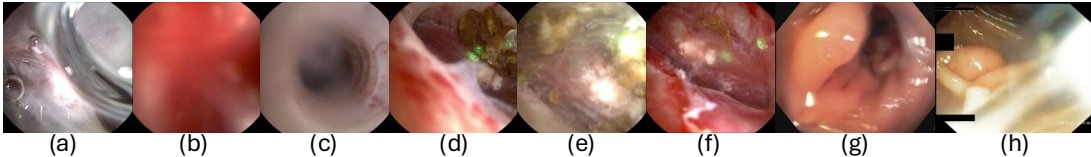

Figure 1: Challenging ureteroscopy (a–f, left) and colonoscopy (g–h, right) images for segmentation. (a) irrigation; (b) bleeding; (c) motion blur; (d) early ablation; (e) mid ablation; (f) late ablation. (g) and (h) are from the public dataset (Ali et al., 2023), which is collected from multiple imaging sites. Enlarged views in Fig. 5

.

itself. A key approach to achieving this is enforcing consistency constraints (Tarvainen and Valpola, 2017), either through uncertainty-guided self-regularization (Sohn et al., 2020; Yang et al., 2023; Luo et al., 2022b; Wang et al., 2024; Tarvainen and Valpola, 2017) or cross-supervision (Chen et al., 2021; Luo et al., 2022a) to improve the quality and reliability of pseudo-labels.

Based on these principles, SSL can be broadly categorized into single-network and dual-network frameworks. Single-network approaches enforce consistency under perturbations and regularize pseudo-labels based on uncertainty. (Sohn et al., 2020; Yang et al., 2023; Wang et al., 2024). However, single model-based method tends to persist in its incorrect predictions, leading to error accumulation. Dual-network approaches maintain two networks that exchange pseudo-labels for cross-supervision (Chen et al., 2021; Luo et al., 2022a) to mitigate confirmation bias (Arazo et al., 2020). Building on this, numerous studies in medical imaging have achieved excellent segmentation performance (Luo et al., 2022a,b; Wang et al., 2023; Yu et al., 2019; Lei et al., 2022).

These existing SSL methods have some limitations: **(1)** Single-network methods lack model-level consistency, which makes them struggle with high-uncertainty samples. **(2)** Methods that either use the entire uncertainty map or apply a fixed uncertainty threshold treat many unreliable regions as confident, leading to false positives and overfitting to incorrect pseudo-labels. **(3)** Cross-supervision methods do not explicitly model uncertainty and struggle to filter out unreliable pseudo-labels. Since each model generates pseudo-labels independently, confirmation bias may occur when both models make similar wrong predictions.

In this paper, we propose **Endo-SemiS**, a semi-supervised segmentation method to address the limitations of existing approaches in endoscopic imaging with robust outcomes. Specifically, to address each of these limitations: **(1)** Endo-SemiS adopts a cross-supervision framework (see Fig. 2(a)) to prevent biased learning (Chen et al., 2022) and uses naive U-Net models to ensure real-time clinical applicability (Wei et al., 2021; Luo et al., 2019) rather than relying on transformer-based models that may require heavy computation (Luo et al., 2022a; Wang et al., 2024). **(2)** To obtain reliable pseudo-labels for unlabeled data, a critical step in SSL (Wu et al., 2021), we leverage both aleatoric and epistemic uncertainty (see Fig. 2(b)). Unlike existing fixed-threshold approaches (Sohn et al., 2020), a dynamic thresholding mechanism is applied per uncertainty map, ensuring that only high-confidence

regions contribute to pseudo-label supervision. **(3)** To achieve accurate and consistent supervision, we introduce a joint pseudo-labeling strategy as shown in Fig. 2(c), where supervision is guided by the predictions in the lowest uncertainty regions identified by both networks, and pixels that are classified as uncertain are excluded. **(4)** We design multi-level mutual learning (see Fig. 2(d)) between networks to further mitigate confirmation bias and improve consistency between networks for producing reliable pseudo-labels. Our main contributions are:

- We propose an uncertainty-guided pseudo-labeling approach within a cross-supervision framework, which dynamically filters out unreliable regions for each image and provides more reliable segmentation supervision from unlabeled endoscopic frames.

- We introduce a consistency-focused learning framework with joint pseudo-label supervision and multi-level mutual learning. The more reliable prediction between the two networks is selected as supervision, while mutual learning reduces unnecessary prediction variance in confident regions and leads to more stable pseudo-labels.

- We design a plug-and-play correction model that uses spatiotemporal information from video to refine segmentation and can be easily integrated into other frameworks.

We validate Endo-SemiS on kidney stone laser lithotripsy as a challenging primary task and on polyp screening across different centers to demonstrate generalizability. Our comprehensive evaluation shows consistent improvements over state-of-the-art methods.

## 2. Methods

We begin with a semi-supervised segmentation dataset $D$, which consists of limited labeled data $\{x_l, y_l\}$ and a large amount of unlabeled data $\{x_u\}$, where $x$ and $y$ represent the input images and their annotations, respectively.

### 2.1. Preliminaries

**Generic pseudo-label learning.** The generic pseudo-label learning (Bellver Bueno et al., 2019) for a single network (referred to as Generic) first trains the model $f$, with forward pass $f(\cdot)$ on $\{x_l, y_l\}$ and applies it to $x_u$ to obtain the logit map $f(x_u)$, which is then binarized to form pseudo-label $\tilde{y}_u$ and used as additional supervision. This can be described as:

$$L = L_s + L_p \tag{1}$$

where $L_s$ and $L_p$ denote the supervised and pseudo-supervised loss for $\{x_l, y_l\}$ and $\{x_u, \tilde{y}_u\}$.

**Cross-supervision.** Endo-SemiS employs two individual U-Nets without sharing weights (Ronneberger et al., 2015) to achieve cross-supervision signals, as shown in Fig. 2(a). For a given input $x \in \{x_l, x_u\}$, the supervision can be simply extended from Generic (Eq. 1) as:

$$L_p^{\text{cross}}(x) = L_p(f_1(x), \tilde{y}_2) + L_p(f_2(x), \tilde{y}_1) \tag{2}$$

where $L_p^{\text{cross}}$ represents the cross-supervision applied to both networks using the pseudo-label from the other model. The subscripts $i \in \{1, 2\}$ indicate the corresponding network. Note that $f_i(x)$ denotes the raw logit map produced by network $i$ for input $x$. For brevity, we include it in the loss function, as it can be converted to probabilities within the loss.

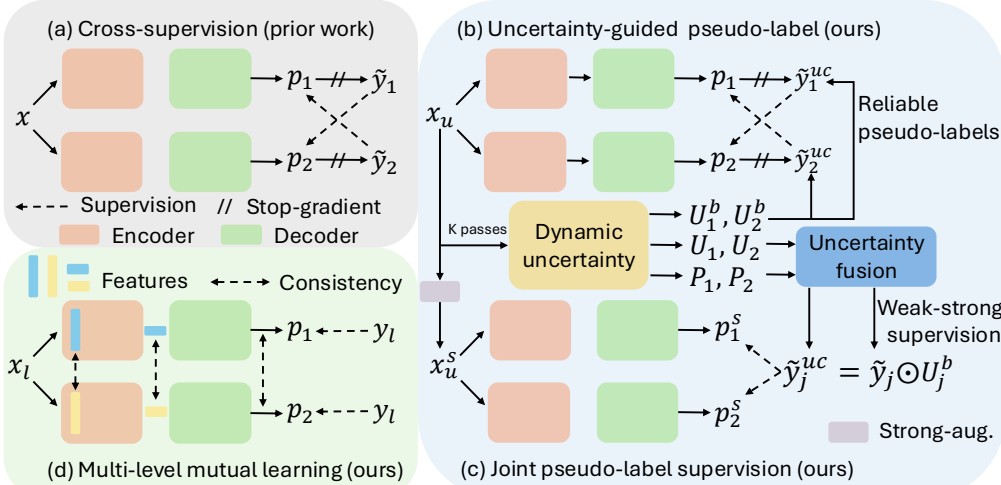

Figure 2: The proposed framework adapts the widely used cross-supervision baseline (a) with uncertainty-guided supervision to obtain reliable pseudo-labels (b–c), and further incorporates multi-level mutual learning (d) to improve cross-network consistency. Panels (b–c) (in blue) operate only on unlabeled data $x_u$, whereas (d) is applied only to labeled data $x_l$. The two networks share the same architecture but are optimized independently. $y$, $\tilde{y}$, and $\tilde{y}^{uc}$ denote the ground-truth mask, the raw pseudo-label, and the uncertainty-guided pseudo-label, respectively. $\odot$ denotes the Hadamard (element-wise) product, and $U^b$ is the binary mask from uncertainty map $U$. $x_u^s$ represents a strongly intensity-augmented version of $x_u$. We define $\tilde{y}_1^{uc} = \tilde{y}_1 \odot U_1^b$ and $\tilde{y}_2^{uc} = \tilde{y}_2 \odot U_2^b$, and omit them for brevity.

## 2.2. Uncertainty-guided pseudo-label

Uncertainty is introduced into the framework to mitigate confirmation bias (Fig. 2(b)). *We hypothesize that uncertainty estimates allow us to identify unreliable pseudo-label regions and exclude them from supervision, so that training focuses on reliable areas.*

**Aleatoric uncertainty.** We adopt the widely used weak-to-strong augmentation strategy (Sohn et al., 2020). Each unlabeled image $x_u$ first undergoes geometric augmentations, referred to as weak augmentation, and $x_u$ is further modified using intensity-based augmentations to obtain a strongly augmented image $x_u^s$. The corresponding pseudo-label $\tilde{y}_u$ is used to supervise the prediction from $x_u^s$. We also leverage CutMix (Yun et al., 2019) augmentation on $x_u$ and $x_u^s$ to further increase the robustness and segmentation performance.

**Epistemic uncertainty.** The cross-supervision setup naturally accommodates stochastic regularization, so we insert Monte Carlo dropout (Kendall and Gal, 2017) layers after each decoder convolution to estimate uncertainty and improve the reliability of pseudo-labels, which further improves segmentation performance (Yu et al., 2019). Specifically, as shown in Fig. 3(a), each unlabeled sample $x_u$ is passed through both networks multiple times to estimate entropy-based uncertainty. For each network $f_i$ ($i \in \{1, 2\}$), the final output

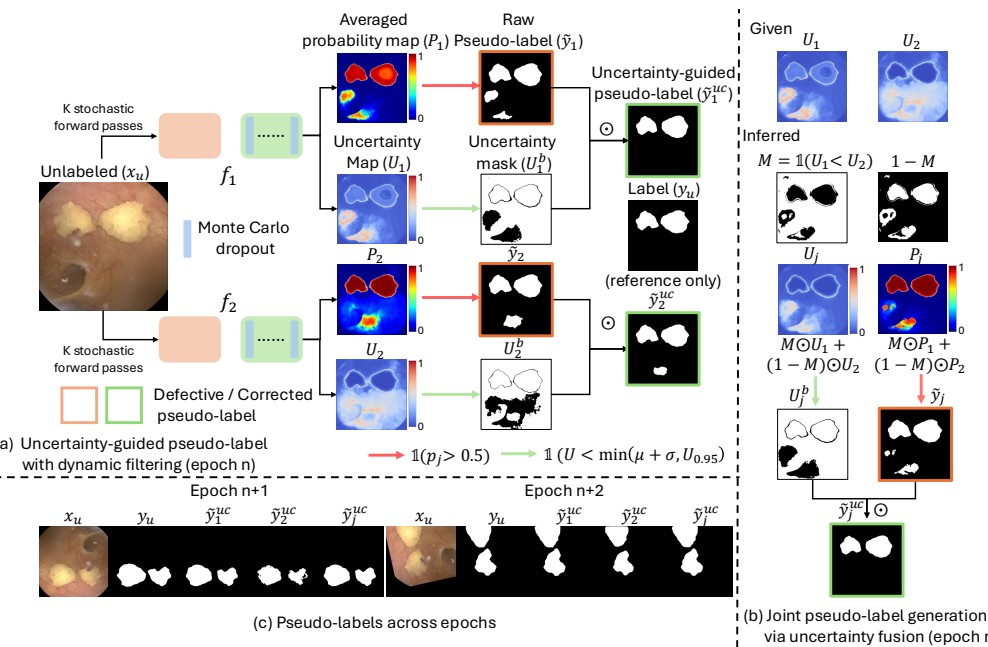

Figure 3: (a) For an unlabeled image $x_u$, uncertainty-guided pseudo-labels $\tilde{y}_1^{uc}$ and $\tilde{y}_2^{uc}$ (green boxes) are obtained by dynamically filtering the raw pseudo-labels $\tilde{y}_1$ and $\tilde{y}_2$, leading to cleaner supervision. The label $y_u$ of the unlabeled image is shown for reference only. (b) $M$ chooses the lower-uncertainty prediction at each pixel to obtain the joint pseudo-label $\tilde{y}_j^{uc}$ for more reliable supervision by correcting residual defects in $\tilde{y}_2^{uc}$ from (a). (c) Compared with the pseudo-labels at epoch $n$ in (a), the $\tilde{y}_1^{uc}$, $\tilde{y}_2^{uc}$ and $\tilde{y}_j^{uc}$ at epochs $n+1$ and $n+2$ become cleaner and more consistent with $y_u$, indicating the effectiveness of (a) and (b).

probability map is computed as $P_i = \frac{1}{K}\sum_{k=1}^{K} p_{i,k}$, where $p_{i,k}$ denotes the probability map in the $k$-th forward pass of network $i$, and we set $K = 5$. The entropy-based epistemic uncertainty map is derived as $U_i = \frac{1}{K}\sum_{k=1}^{K} h(p_{i,k})$, with $h(p) = -p\log p - (1-p)\log(1-p)$.

**Dynamic filtering.** Unlike previous works that use a fixed threshold (Sohn et al., 2020), the entire uncertainty map (Luo et al., 2022b) or quantile-based selection (Yu et al., 2019; Yang et al., 2023), we use a dynamic and data-driven thresholding strategy. Given $U_i$, the threshold is set as $T_i = \min[\mu(U_i) + \sigma(U_i), U_{i,0.95}]$, where $\mu$, $\sigma$ and $U_{i,0.95}$ denote the mean, standard deviation and $95^{th}$ percentile, respectively. Our adaptive thresholding effectively handles long-tail distributions and noisy predictions, yielding a more reliable uncertainty-based binary mask $U_i^b = \mathbb{1}(U_i < T_i)$, where $\mathbb{1}$ denotes the indicator function (see Fig. 3(a)). The final uncertainty-guided pseudo-label for $x_u$ is then formulated as $\tilde{y}_i^{uc} = \tilde{y}_i \odot U_i^b$.

## 2.3. Joint pseudo-label supervision

Even with the incorporation of uncertainty estimates, the pseudo-labels may still be too noisy to provide appropriate supervision for harder samples. Most existing methods solely

rely on the $\tilde{y}_u$ from each network for supervision, which may not be sufficient. To address this, *our hypothesis is that joint supervision can effectively refine pseudo-labels by leveraging complementary information from both networks, providing more reliable supervision for challenging samples.*

As shown in Fig. 3(b), the joint pseudo-label $\tilde{y}_j^{uc}$ is constructed in three steps: (1) Given the uncertainty maps $U_1$ and $U_2$ from the two networks in Endo-SemiS, we create a binary mask $M = \mathbb{1}(U_1 < U_2)$ that selects the more confident prediction at each pixel. (2) Using this mask, we form the joint probability $P_j = M \odot P_1 + (1 - M) \odot P_2$ and obtain the raw pseudo-label $\tilde{y}_j$ by thresholding $P_j$ at 0.5, while the joint uncertainty map is defined as $U_j = M \odot U_1 + (1 - M) \odot U_2$. (3) Finally, we apply the dynamic filtering scheme to $U_j$ to obtain the binary uncertainty mask $U_j^b$ and compute the final uncertainty-guided joint pseudo-label as $\tilde{y}_j^{uc} = \tilde{y}_j \odot U_j^b$.

For an unlabeled image $x_u$ and its strongly augmented version $x_u^s$, we extend the cross-supervision loss in Eq. 2 to a weak–strong setting, where pseudo-labels are generated from the weak augmented image (see Sec. 2.2) and used to supervise the strongly augmented image. Together with uncertainty-guided pseudo-label learning, the cross pseudo-supervised loss $L_p^{\mathrm{cross}}(x_u, x_u^s)$ is defined as:

$$L_p^{\mathrm{cross}}(x_u, x_u^s) = \underbrace{L_p\big(f_1(x_u), \tilde{y}_2^{uc}\big) + L_p\big(f_2(x_u), \tilde{y}_1^{uc}\big)}_{\text{uncertainty-guided cross-supervision}} + \underbrace{L_p\big(f_1(x_u^s), \tilde{y}_j^{uc}\big) + L_p\big(f_2(x_u^s), \tilde{y}_j^{uc}\big)}_{\text{joint pseudo-label supervision}} \quad (3)$$

## 2.4. Multi-level mutual learning

Individual networks may independently learn different representations, which can cause divergence and inconsistencies in their predictions. If one network is consistently wrong, it can bias the other network and propagate errors. We propose a multi-level mutual learning approach to mitigate this variability by aligning the learning trajectories of both models and promoting consistency in their predictions. Although it does not guarantee correctness on unlabeled data, it reduces randomness and stabilizes the learning process, making models less likely to reinforce extreme errors.

We use the labeled data to apply mutual learning between the two networks. This encourages similarity at both the encoders and the decoders. The consistency from encoder and bottleneck features helps align feature representations and reduce variability in learned embeddings. Unlike previous work, which enforces the similarity between the probability maps (Zhang et al., 2018), we enforce prediction consistency at the decoder level by aligning the logit maps of the networks, which is particularly important when generating pseudo-labels. Since pseudo-labels are filtered based on confidence thresholds, mutual learning stabilizes training by reducing prediction variance between networks, making the pseudo-label selection process more reliable.

For a labeled image $x_l$, let $f_1^e, f_1^b, f_1^l$ and $f_2^e, f_2^b, f_2^l$ denote the first encoder feature maps, bottleneck features, and logit maps of the two networks, respectively. The multi-level mutual learning loss is defined as:

$$L_m(x_l) = L_{\mathrm{ssim}}\big(f_1^e, f_2^e\big) + 0.5 \left(L_{\mathrm{kl}}(p_1^b \parallel p_2^b) + L_{\mathrm{kl}}(p_2^b \parallel p_1^b)\right) + 2\, L_{\mathrm{mse}}\big(f_1^l, f_2^l\big) \quad (4)$$

where $p_i^b = \mathrm{softmax}(f_i^b)$ denotes the channel-wise probability distribution of the bottleneck feature map, $i \in \{1, 2\}$.

**Total objective function.** For labeled and unlabeled data, the total objectives are:

$$L(x_l) = L_s(x_l) + 0.5\, L_p^{\text{cross}}(x_l) + 0.5\, L_m(x_l), \quad L(x_u) = 0.5\, L_p^{\text{cross}}(x_u, x_u^s) \tag{5}$$

### 2.5. Spatiotemporal (ST) correction at frame level

Segmentations produced on semi-supervised frames may exhibit frame-level inconsistencies due to the lack of temporal information, which appear as isolated false positive (FP) or false negative (FN) frames. As a post-processing step, we leverage the inherent spatiotemporal information in video clips, and introduce a separate correction model ($f_{st}$) at frame level to mitigate false positive FP and FN frames.

We denote the $n^{th}$ test frame by $x_n$ and its predicted binary segmentation mask by $\tilde{y}_n$. For each frame $x_n$, we define $R_n$ as the total number of foreground pixels in $\tilde{y}_n$. Our key assumption is that adjacent frames should not exhibit large discrepancies in $R_n$. In particular, for FN frames, the target regions overlap across these frames, whereas for FP frames, the background region remains consistent (or contains little foreground). These assumptions motivate our inter-frame FP/FN detection and correction. We enforce temporal consistency by correcting FP frames when $R_n > 0$ and $R_{n-1} = R_{n+1} = 0$. Similarly, we classify $x_n$ as a FN frame when $R_n = 0$ and $R_{n-1} > r$ and $R_{n+1} > r$. We set $r = \frac{1}{4}HW$, where $H$ and $W$ denote the frame height and width.

To refine the predictions, we train a separate correction model $f_{st}$ that operates on a local temporal window. Given labeled training pairs $\{(x_{n-2}, y_{n-2}), \ldots, (x_{n+2}, y_{n+2})\}$ sampled from $\{x_l, y_l\}$, we concatenate them along the channel dimension to form $c_n$, and use this as input to predict a refined segmentation for the central frame $x_n$. During training, random corruptions are introduced to the masks with basic morphological operations or by setting them to zero. We use the MSE loss to enforce spatiotemporal consistency, and the total loss is:

$$L = L_s(f_{st}(c_n), y_n) + 0.25 \sum_{k \in \{-1,1\}} L_{\text{mse}}(f_{st}(c_n), y_{n+k}) + 0.1 \sum_{k \in \{-2,2\}} L_{\text{mse}}(f_{st}(c_n), y_{n+k}) \tag{6}$$

This formulation allows the network to leverage spatiotemporal information while preventing it from overly dominating the training process, thereby accommodating potential variations between frames. For inference, the correction model $f_{st}$ is applied to frames classified as FP or FN, and uses adjacent masks to satisfy the local-consistency assumption for challenging ureteroscopy videos.

## 3. Experiments

**Kidney stone dataset.** This in-house dataset (Deol et al., 2024) consists of 38 fiberoptic and 98 digital endoscopy videos. We extracted frames at 3 FPS, resulting in a total of 21,718 labeled frames. We partitioned the data at the video-level, yielding approximately a 75/5/20% split for training/validation/testing. While all videos contain kidney stones, some individual frames may not, which introduces an implicit detection challenge in addition to segmentation. The dataset exhibits substantial variation in image quality due to the complex in vivo surgical environment (Fig. 1, Appendix. A), such as rapid motion, debris and fluctuating lighting conditions. All images are resized to $256{\times}256$. Detailed information about the problem setting and key challenges is provided in Appendix A.

**Polyp colonoscopy dataset.** PolypGen (Ali et al., 2023) is a public colonoscopy dataset collected from six imaging centers. It contains 1,537 single-labeled frames, which are discretely sampled and focus on polyp-present images, and 2,225 sequence-labeled frames sampled from short video clips, which may include both polyp-present and polyp-absent views. The sequence setting is more challenging due to larger appearance variation, motion blur, and frequent polyp-absent frames. Following the benchmark (Ali et al., 2023), we train on frame data from centers 1-5 and evaluate on center 6 for both frame and sequence data. All images are resized to $512 \times 512$. Detailed information is provided in Appendix A.

**Implementation details.** During training, we set the $L_s$ and $L_p$ as naive binary cross entropy loss with a batch size of 16 for 200 epochs. The initial learning rate is $10^{-4}$ with a cosine curve decay to $10^{-5}$. Our study was conducted on an NVIDIA A6000.

**Compared methods.** We compare to several state-of-the-art semi-supervised learning methods, including Generic (Bellver Bueno et al., 2019), AllSpark (Wang et al., 2024), UPRC (Luo et al., 2022b), FixMatch (Sohn et al., 2020), UniMatch (Yang et al., 2023), Mean Teacher (Tarvainen and Valpola, 2017), Cross-Pseudo Supervision (CPS) (Chen et al., 2021) and Cross Teaching (Luo et al., 2022a). For polyp datset, we additionally compare state-of-the-art polyp segmentation methods (PNS+ (Ji et al., 2022) and DSHNet (Wang et al., 2025)) as well as a lightweight CNN (EfficientNet (Tan and Le, 2019)). Further details are provided in Appendix B.

**Evaluation metrics.** We report pixel-level segmentation performance using Dice, sensitivity, and specificity. We also evaluate image-level target presence detection by converting each predicted mask into a binary image label. An image is predicted positive if any foreground pixel is present and negative otherwise. The precision, recall, F1-score, and accuracy are computed at the image level. These metrics indicate whether the model detects the presence or absence of the target object, independent of pixel-wise overlap quality.

**Segmentation performance.** The quantitative results of the kidney stone dataset using 10% labeled data are shown in Tab. 1. The Generic model underperforms compared to supervised learning, which highlights the critical role of pseudo-label quality in semi-supervised segmentation. In contrast, the results of Mean Teacher, UniMatch, and FixMatch show that incorporating external uncertainty improves segmentation, especially for UniMatch where epistemic uncertainty is also leveraged. The results of AllSpark indicate that transformer-based method struggles for kidney stone segmentation, where image quality is variable (Fig. 4, enlarged viewed in Fig. 7). Cross-supervision methods (lavender) achieve better performance than single-network-based methods (blue), demonstrating better generalizability. Endo-SemiS achieves substantially superior performance across most metrics compared to these SOTA semi-supervised methods. Notably, it even outperforms supervised methods trained on full labeled data (upper bound, green).

**Consistency analysis.** In Tab. 2, we present consistency results in two aspects: (1) robustness across different ratios of labeled training data, and (2) consistency between models within the framework. Endo-SemiS maintains stable performance across different ratios, demonstrating particularly robust performance when labeled data is extremely limited (only 1%). The performance of the two cross-supervised models of our framework

Table 1: Kidney results ($mean \pm stdev.$, in %) with **10% labeled data**. Bold indicates the **best**. The horizontal sections show: supervised (gray), semi-supervised with single network (blue), cross-supervised (lavender), and supervised with all labeled data, i.e., upper bound (green). Our method achieved the highest Dice score, sensitivity, F1, and accuracy. The compared methods are described in Appendix B.

| | Pixel-level | | | Image-level | | | |
|---|---|---|---|---|---|---|---|
| Methods | Dice | Sensitivity | Specificity | Pre. | Rec. | F1 | Acc. |
| U-Net | 80.5±32.1 | 88.6±22.0 | 95.4±8.4 | 88.7 | 95.3 | 92.8 | 90.1 |
| nnU-Net | 79.5±33.8 | 85.9±27.4 | 95.5±9.1 | 90.1 | 91.1 | 90.6 | 87.6 |
| Generic | 78.5±31.7 | 86.1±25.7 | 92.3±13.9 | 90.7 | 95.3 | 92.9 | 90.5 |
| AllSpark | 77.0±31.2 | 88.0±24.8 | 89.3±18.0 | 94.7 | 92.8 | 93.8 | 91.7 |
| UPRC | 80.7±31.4 | 84.0±27.3 | 96.4±7.8 | 92.9 | 94.6 | 93.7 | 91.6 |
| FixMatch | 81.9±31.7 | 89.8±22.4 | 94.3±10.9 | 89.7 | **96.5** | 93.0 | 90.5 |
| UniMatch | 85.5±27.6 | 89.4±23.2 | 95.5±8.9 | 94.3 | 96.4 | 95.4 | 91.7 |
| Mean Teacher | 82.2±31.2 | 84.1±28.6 | 96.6±8.5 | 95.6 | 90.5 | 93.0 | 91.1 |
| CPS | 85.2±28.0 | 88.8±22.8 | 95.8±8.8 | 94.0 | 96.1 | 95.0 | 93.4 |
| Cross Teaching | 85.6±28.7 | 87.6±26.5 | **96.7±7.4** | **96.5** | 92.6 | 94.8 | 92.9 |
| Endo-SemiS (Ours) | **87.6±26.4** | **91.1±21.5** | 96.0±8.4 | 95.0 | 96.1 | **95.6** | **94.1** |
| Upper bound U-Net | 85.3±29.2 | 89.0±24.5 | 96.5±8.2 | 94.4 | 94.2 | 94.3 | 92.5 |
| Upper bound nnU-Net | 85.5±28.5 | 89.3±24.5 | 96.0±8.6 | 92.4 | 93.3 | 92.9 | 90.5 |

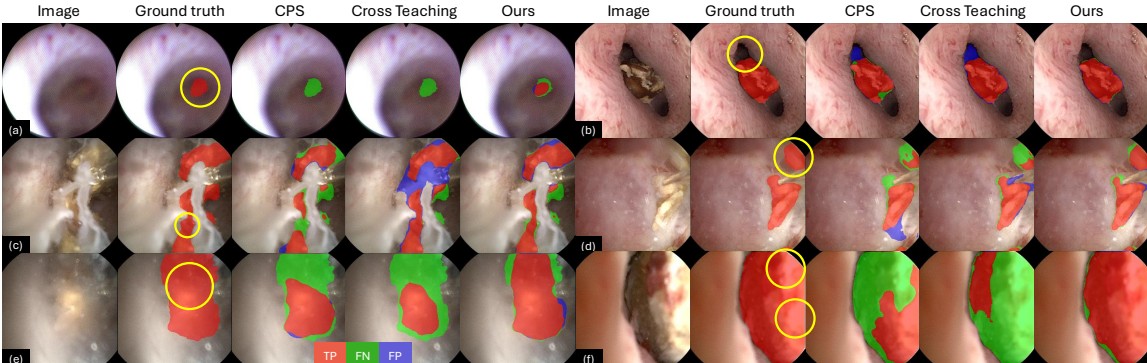

Figure 4: Qualitative kidney stone results (10% labeled data). Yellow circles highlight poor visibility areas. Enlarged, high-resolution views are shown in Fig. 7.

is more consistent and reliable than the compared methods. Considering the challenging visibility conditions in kidney stone surgery (Fig. 4), consistency is crucial to performance because inaccurate pseudo-labels can severely degrade segmentation results. Finally, we observe that our ST corrective model improves performance across all label ratios.

Table 2: Dice (%) on kidney dataset with various labeled data ratios. "-1" and "-2" denote individual networks for cross-supervision. ST: spatiotemporal correction. Bold indicates the best in each category. Lavender denotes the cross-supervised methods.

| Methods | 1% | 5% | 10% | 30% | 100% |
|---|---|---|---|---|---|
| U-Net | 74.9±34.1 | 77.8±34.5 | 80.5±32.1 | 82.0±32.0 | 85.3±29.2 |
| nnU-Net | 76.4±34.3 | 78.0±34.5 | 79.5±33.8 | 82.1±31.6 | 85.5±28.5 |
| Generic | 69.4±37.3 | 76.5±34.3 | 78.5±31.7 | 83.4±29.6 | - |
| CPS-1 | 82.9±30.5 | 84.7±28.8 | 85.2±28.0 | 85.7±27.7 | - |
| Cross Teaching-1 | 77.1±32.4 | 80.1±32.2 | 85.6±28.7 | 86.5±27.6 | - |
| Endo-SemiS-1 | **86.5±27.6** | **87.5±26.4** | **87.6±26.4** | **87.9±26.1** | - |
| CPS-1+ST | 83.8±29.5 | 85.3±28.1 | 85.7±27.4 | 86.2±27.1 | - |
| Endo-SemiS-1+ST | **87.1±27.1** | **87.8±26.3** | **88.1±25.7** | **88.2±25.8** | - |
| Performance variability in cross-supervised segmentation (±Dice in %) | | | | | |
| CPS-2 | -1.0 | +1.9 | -0.7 | +0.6 | - |
| Cross Teaching-2 | -11.4 | -13.6 | -4.0 | -4.6 | - |
| Endo-SemiS-2 | **-0.9** | **+0.1** | **0.0** | **-0.2** | - |

Table 3: Dice (%) for ablation study on kidney dataset with 10% labeled data. AU, EU: aleatoric/epistemic uncertainty. JPS: joint pseudo-label supervision. ML-D: mutual learning in decoder. ML-EB: mutual learning in encoder and bottleneck.

| | Baseline (Chen et al., 2021) | + AU | + EU | + JPS | + ML-D | + ML-EB |
|---|---|---|---|---|---|---|
| Endo-SemiS-1 | 85.2 | 86.2 | 86.9 | **87.8** | 87.2 | 87.6 |
| Endo-SemiS-2 | 84.5 | 86.4 | 87.2 | 86.8 | 87.5 | **87.6** |

**Ablation analysis.** Tab. 3 shows the ablation study, where CPS is used as the baseline method, and the improvements for each added component are shown. Importantly, joint pseudo-label supervision (JPS) yields a larger improvement, which indicates that it effectively removes uncertain regions and generates high-quality pseudo-labels for supervision, especially for strong augmented images. Although multi-level mutual learning slightly decreases the performance, it improves consistency.

**Kidney stone size analysis.** We stratify the test set (n=3959) by kidney stone size using the ground truth mask area relative to the image area. Table. 4 shows that Endo-SemiS achieves the best semi-supervised Dice in the Small/Medium/Large groups and the best overall Dice. Among SSL methods, the largest improvement is for large stones. In intra-operative settings, close-range views are common and often involve ongoing ablation, which increases boundary ambiguity and makes segmentation more challenging (Fig. 5). Compared to cross-supervised methods, our method still shows a clear improvement on

Table 4: Dice score (%) stratified by kidney stone size. Size categories are defined by the ground truth mask area relative to the image area ($H \times W$): No Stone (0), Small ($\leq \frac{1}{8}HW$), Medium ($\frac{1}{8}HW - \frac{1}{4}HW$), and Large ($\geq \frac{1}{4}HW$). $n$ denotes the number of samples. Bold indicates the **best**. Lavender denotes the cross-supervised methods. Endo-SemiS achieves the best overall Dice with 10% labels.

| Method | Label usage | No Stone (n=1364) | Small (n=495) | Medium (n=418) | Large (n=1682) | Overall |
|---|---|---|---|---|---|---|
| U-Net | 100% | 91.3±29.6 | 63.8±38.5 | 82.0±25.9 | 87.6±22.8 | 85.3±29.2 |
| nnU-Net | 100% | 90.2±29.1 | 64.9±38.9 | 82.2±25.3 | 88.6±21.6 | 85.5±28.5 |
| U-Net | 10% | 75.4±42.5 | 65.6±35.2 | 82.9±20.5 | 88.4±16.8 | 80.5±32.1 |
| nnU-Net | 10% | 80.9±39.4 | 57.6±40.6 | 77.4±29.6 | 85.3±23.7 | 79.5±33.8 |
| UniMatch | 10% | 89.0±31.3 | 69.2±35.7 | 81.4±25.0 | 88.5±19.3 | 85.5±27.6 |
| CPS | 10% | 88.3±32.2 | 65.7±37.0 | 83.1±22.0 | 88.9±18.6 | 85.2±28.0 |
| Cross Teaching | 10% | **93.5±24.6** | 65.9±38.6 | 83.8±26.0 | 87.1±24.3 | 85.6±28.7 |
| Endo-SemiS (Ours) | 10% | 90.5±29.4 | **70.4±36.0** | **84.4±23.0** | **91.1±17.9** | **87.6±26.4** |

small stones, suggesting better robustness on challenging small-region segmentation, where limited pixel support makes predictions sensitive to noise. Overall, these results show that the proposed uncertainty-guided learning remains effective across stone sizes. However, the "No Stone" group Dice suggests occasional false positives.

**Generalizability analysis.** We further evaluate Endo-SemiS on the PolypGen cross-center setting (Tab. 5) to explicitly assess robustness to domain shift.

With only 10% labeled data, key findings are: (1) supervised training can perform well on single-frame evaluation but often degrades noticeably on sequence evaluation in the cross-center setting, indicating limited robustness when models are trained on frames but tested on sequences under domain shift with substantial appearance changes. (2) Using the same U-Net backbone, Endo-SemiS consistently improves over supervised training and SSL methods on both single-frame and sequence evaluation, demonstrating stronger generalization despite these appearance differences across domains, and the detailed uncertainty analysis can be viewed in Appendix C. (3) While a stronger backbone improves single-frame performance, sequence predictions can become less stable under domain shift. Endo-SemiS alleviates this issue with more reliable sequence predictions. (4) Endo-SemiS is model-agnostic and can benefit from a stronger backbone without sacrificing its single-frame advantages, while mitigating its weaknesses on sequence evaluation by producing more stable predictions. (5) A lightweight CNN (EfficientNet) shows limited performance under supervised training with limited labels. Yet when trained with Endo-SemiS, it achieves competitive sequence performance comparable to heavier backbones while exhibiting the smallest gap between single-frame and sequence evaluation. This further validates the model-agnostic property of our methods.

Table 5: Quantitative results in polyp dataset with 10% labeled data. **Green** and **yellow** denote the best and second best mean Dice, respectively. Lavender denotes the cross-supervised methods. Endo-SemiS is model-agnostic and shows strong performance across diverse backbones with only 10% labeled data, particularly on sequence data that better reflects real-world endoscopic procedures.

| Methods | Single frame data | | | Sequence frame data | | |
|---|---|---|---|---|---|---|
| | Dice | Sensitivity | Specificity | Dice | Sensitivity | Specificity |
| U-Net | 75±30 | 73±31 | 100±1 | 64±38 | 64±38 | 100±1 |
| nnUNet | 75±33 | 78±32 | 99±3 | 53±43 | 68±40 | 99±4 |
| PNS+ | 71±34 | 71±35 | 99±2 | 48±41 | 47±42 | 99±3 |
| DSHNet | **80±24** | 87±24 | 99±1 | 59±39 | 76±31 | 99±2 |
| EfficientNet | 67±33 | 69±35 | 99±2 | 44±39 | 41±37 | 99±2 |
| CPS-1 (UNet) | 77±32 | 74±33 | 100±1 | 68±40 | 65±40 | 100±1 |
| CPS-2 (UNet) | 75±34 | 71±34 | 100±1 | 64±42 | 65±42 | 100±1 |
| Cross Teaching-1 (UNet) | 73±35 | 70±36 | 100±1 | 63±41 | 60±41 | 100±1 |
| Cross Teaching-2 (UNet) | 75±34 | 73±34 | 100±1 | 65±40 | 63±40 | 100±1 |
| Endo-SemiS-1 (UNet) | 76±34 | 75±34 | 100±1 | 69±39 | 67±39 | 100±1 |
| Endo-SemiS-2 (UNet) | 79±30 | 77±31 | 100±1 | **71±37** | 70±37 | 100±2 |
| Endo-SemiS-1 (DSHNet) | **80±28** | 82±29 | 100±1 | **73±34** | 73±34 | 99±1 |
| Endo-SemiS-2 (DSHNet) | 78±30 | 82±29 | 99±1 | **73±35** | 73±34 | 100±1 |
| Endo-SemiS-1 (EfficientNet) | 73±31 | 81±32 | 99±2 | 70±36 | 73±35 | 99±2 |
| Endo-SemiS-2 (EfficientNet) | 74±32 | 79±33 | 99±2 | **71±36** | 71±36 | 99±2 |
| Upper bound U-Net | 79±30* | 79±31 | 99±2 | 69±37* | 74±35 | 99±2 |
| Upper bound nnUNet | **80±29** | 84±26 | 99±1 | 62±41 | 71±38 | 99±2 |
| Upper bound PNS+ | 74±32 | 77±33 | 99±3 | 49±42 | 49±43 | 99±3 |
| Upper bound DSHNet | **85±24** | 92±19 | 99±2 | 66±39 | 79±33 | 99±3 |
| Upper bound EfficientNet | 75±32 | 77±34 | 99±1 | 45±40 | 41±40 | 100±1 |

* denotes our implementation; benchmark (Ali et al., 2023) results are 79% and 66%.

## 4. Conclusion

In this study, we propose **Endo-SemiS** for robust endoscopic segmentation using semi-supervised learning. It uses an uncertainty-guided pseudo-label strategy, cross- and joint-supervision, and mutual learning, and achieves strong performance on two endoscopy datasets with substantial variations in image quality. Endo-SemiS can still fail when the model produces confidently wrong predictions that pass the uncertainty filter (Appendix D), and we observe residual false positives in stone-absent frames (Tab. 4), reflecting the coupled detection-segmentation challenge in videos that contain negative frames despite stone-positive videos. Future work will validate Endo-SemiS on additional datasets (e.g., SunSeg (Ji et al., 2022)) and further incorporate temporal information into the semi-supervised learning framework.

## Acknowledgments

This work was supported in part by the National Institutes of Health (R21DK133742) and Vanderbilt Institute for Surgery and Engineering (VISE) Seed Grant. Daiwei Lu is supported by NIH F31DK143735-01.

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

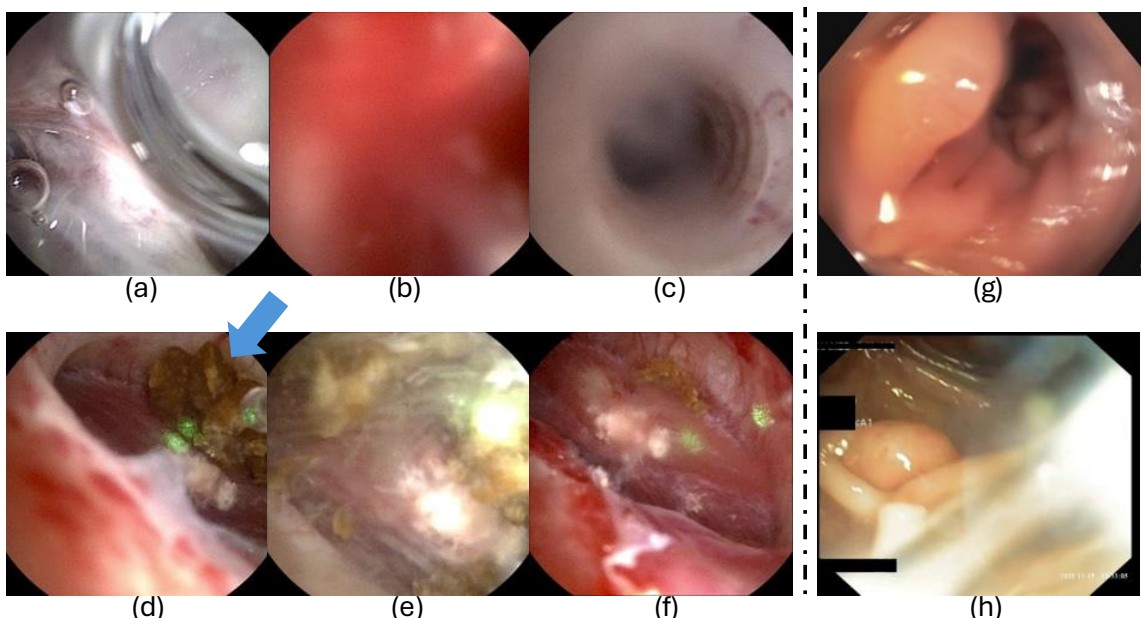

Figure 5: Challenging ureteroscopy (a–f, left) and colonoscopy (g–h, right) images for segmentation. (a) irrigation; (b) bleeding; (c) motion blur; (d) early ablation; (e) mid ablation; (f) late ablation. The arrow indicates the target kidney stone for ablation. (g) and (h) are from the public dataset (Ali et al., 2023), which is collected from multiple imaging sites.

## Appendix A. Dataset Challenges

Fig. 5 provides representative challenging examples from both datasets. For the kidney stone ureteroscopy data (a–f), the in vivo surgical environment introduces strong image-quality degradation, including irrigation flow, bleeding, rapid camera motion, and ablation-induced debris/bubbles and specular highlights. Panels (d-f) illustrate a typical **localization-to-ablation** workflow: (d) the surgeon first **locates** the target stone (arrow), (e) the scope then moves closer and **ablation begins**, and (f) shows the view **after ablation**. During this transition, the stone may remain only partially visible and become increasingly affected by blood, movements, debris, and lighting changes as the scope approaches, resulting in ambiguous boundaries and occasional stone-absent frames despite stone-positive videos. For the PolypGen colonoscopy data (g-h), cross-center acquisition yields noticeable appearance shifts (color/texture/illumination), and the sequence clips often contain motion blur and frequent polyp-absent frames, which effectively couples segmentation with an implicit detection challenge. Overall, the figure summarizes the main sources of difficulty that our method targets: cross-site appearance variation and severe, procedure-specific artifacts.

## Appendix B. Compared methods

We compare our method with several state-of-the-art semi-supervised approaches (Wang et al., 2024; Sohn et al., 2020; Yang et al., 2023; Tarvainen and Valpola, 2017; Luo et al., 2022b; Chen et al., 2021; Luo et al., 2022a). These methods cover both single-network (Wang et al., 2024; Sohn et al., 2020; Yang et al., 2023; Tarvainen and Valpola, 2017) and cross-supervision (Chen et al., 2021; Luo et al., 2022a) frameworks, with and without transformer backbones (Luo et al., 2022a; Wang et al., 2024). They focus on different uncertainty modeling strategies, including aleatoric (Wang et al., 2024; Sohn et al., 2020; Yang et al., 2023) and epistemic (Yang et al., 2023; Tarvainen and Valpola, 2017; Luo et al., 2022b) uncertainty, and combine confidence-based pseudo-labeling (Sohn et al., 2020; Yang et al., 2023; Chen et al., 2021; Luo et al., 2022a; Wang et al., 2024) with uncertainty-guided self-consistency (Tarvainen and Valpola, 2017; Luo et al., 2022b). For completeness, we summarize the main characteristics of each method below.

- **AllSpark** (Wang et al., 2024): Single-network transformer-based semi-supervised semantic segmentation method built on a standard pseudo-labeling baseline. It inserts an AllSpark bottleneck between the encoder and decoder, where channel-wise cross-attention and a class-wise semantic memory reconstruct labeled features from unlabeled features to strengthen supervision. It was published at *CVPR* 2024.

- **Uncertainty-Rectified Pyramid Consistency (URPC)** (Luo et al., 2022b): It is a single-network pyramid-prediction framework for semi-supervised medical image segmentation. The model produces multi-scale predictions and, for unlabeled data, enforces consistency between each scale and their average prediction. Pixel-wise uncertainty is estimated from the discrepancy among scales in a single forward pass and is used both to weight the pyramid consistency loss and to impose an uncertainty-minimization regularizer, enabling more reliable use of unlabeled images. It was published at *Medical Image Analysis* 2022.

- **FixMatch** (Sohn et al., 2020): Single-network method with a CNN backbone that combines consistency regularization and pseudo-labeling. For each unlabeled image, it takes the prediction on a weakly augmented view, keeps it as a qulified pseudo-label only if its confidence exceeds a fixed threshold, and trains the model to match this pseudo-label on a strongly augmented view of the same image. It was published at *NeurIPS* 2020.

- **UniMatch** (Yang et al., 2023): Single-network method with a CNN backbone that revisits FixMatch for semi-supervised semantic segmentation. It maintains weak-strong consistency using fixed confidence-thresholded pseudo-labels from the weakly augmented image, and introduces unified perturbations that operate at both the image level (strong augmentations) and the feature level (dropout), together with two strongly augmented images guided by the same weak prediction, to better exploit the perturbation space. It was published at *CVPR* 2023.

- **Mean Teacher** (Tarvainen and Valpola, 2017): Teacher-Student framework with a single concolutional neural network (CNN) backbone. The student is trained on

labeled data, and an exponential moving average (EMA) of the student weights defines the teacher. For unlabeled data, a consistency loss enforces that the student prediction matches the teacher prediction under stochastic perturbations. This can be viewed as reducing epistemic uncertainty. It was published at *NeurIPS* 2017.

- **Cross Pseudo Supervision (CPS)** (Chen et al., 2021): Cross-supervision semi-supervised semantic segmentation framework in which two segmentation networks with the same architecture but different initializations are trained jointly. For both labeled and unlabeled images, the prediction from each network is used as a pseudo label to supervise the other, enforcing prediction consistency and effectively expanding the training data. It was published at *CVPR* 2021.

- **Cross Teaching between CNN and Transformer (Cross Teaching)** (Luo et al., 2022a): Cross-supervision semi-supervised segmentation framework that pairs a CNN (UNet) and a Transformer (Swin-UNet). On unlabeled images, each network takes the prediction from the other network as a pseudo-label and is optimized with a cross-teaching Dice loss, providing implicit consistency while exploiting the complementary local and long-range representations of CNNs and transformers. It was published at *MIDL* 2022.

For polyp segmentation task, we also compare with fully-supervised methods that address complementary challenges: temporal modeling for video data (PNS+ (Ji et al., 2022)), frequency-domain feature learning (DSHNet (Wang et al., 2025)), and efficient backbone design (EfficientNet (Tan and Le, 2019)).

- **PNS+**: (Ji et al., 2022) It is a video polyp segmentation method that models both long-term and short-term spatial-temporal dependencies via a global-to-local learning strategy. It employs a global encoder to extract anchor frame features and a local encoder to process consecutive frames within a sliding window, with two normalized self-attention (NS) blocks progressively refining spatial-temporal representations. The NS block uses channel splitting, query-dependent relevance measuring, and layer normalization to efficiently capture neighborhood correlations across frames. It was published at *Machine Intelligence Research*, 2022.

- **DSHNet**: (Wang et al., 2025) It is a dynamic spectrum-driven hierarchical learning network for polyp segmentation. It decomposes images into high-frequency and low-frequency components via Discrete Cosine Transform. Specifically, high-frequency features enhance boundary details through skip connections, while low-frequency features guide the generation of dynamic convolution kernels for region-level saliency modeling. The method divides images into polyp interior, boundary, and background regions, applying region-specific kernels to handle polyp heterogeneity. It was published at *Medical Image Analysis* 2025.

- **EfficientNet**: (Tan and Le, 2019) It proposes a compound scaling method that uniformly scales network depth, width, and resolution using a set of fixed scaling coefficients. Starting from a baseline network discovered via neural architecture search

Table 6: Sensitivity analysis on uncertainty estimation hyperparameters. Results report Dice scores (%) on PolypGen (10% labeled). $K$: MC dropout passes. Dynamic thresholding yields consistently strong Dice scores in both the single-frame and sequence settings.

| Threshold | $K$ | Single frame data | | Sequence frame data | |
|---|---|---|---|---|---|
| | | EndoSemiS-1 | EndoSemiS-2 | EndoSemiS-1 | EndoSemiS-2 |
| $\mu + \sigma$ | 5 | 76.1 | 79.1 | 69.8 | 68.9 |
| $P_{95}$ | 5 | 78.3 | 77.4 | 69.6 | 70.1 |
| $\min(\mu + \sigma, P_{95})$ | 5 | 76.2 | **79.4** | 69.3 | 71.2 |
| $\min(\mu + \sigma, P_{95})$ | 10 | 77.3 | 75.0 | **72.7** | 70.6 |
| $\min(\mu + \sigma, P_{95})$ | 15 | 76.0 | 74.8 | 70.5 | 69.8 |

(NAS), it achieves superior accuracy-efficiency trade-offs compared to previous ConvNets. The compound scaling strategy enables more balanced resource allocation across different network dimensions than single-dimension scaling approaches. It was published at *ICML*, 2019.

## Appendix C. Dynamic Uncertainty Thresholding Analysis

Table 6 investigates the sensitivity of Endo-SemiS to the uncertainty-mask thresholding rule and the number of MC-dropout passes $K$ on PolypGen (10% labeled). The two fixed thresholds capture complementary behaviors. The $\mu + \sigma$ rule is more conservative and prioritizes reliability by filtering high-uncertainty pixels, while the percentile rule $P_{95}$ is often more permissive and retains more pixels, which can increase pseudo-label coverage. Our method combines these two fixed choices via a dynamic threshold

$$T = \min\big(\mu(U) + \sigma(U), P_{95}(U)\big),$$

which automatically selects the stricter criterion per image. This inherits the stability of $\mu + \sigma$ when uncertainty is high and still preserves the coverage benefit of $P_{95}$ when the uncertainty distribution is well-behaved, yielding a better reliability-coverage balance for pseudo labels. Consistent with this design, the combined rule achieves the best overall results in Table 6, including the top single-frame Dice for EndoSemi-S-2 (79.4) and strong sequence performance. Increasing $K$ from 5 to 10 can further improve sequence Dice not for frame Dice. While larger $K$ does not provide consistent additional improvements, suggesting that moderate $K$ is sufficient.

## Appendix D. Failure Case Analysis

Fig. 6 visualizes representative failure modes of our uncertainty-guided pseudo-labeling under severe intra-operative appearance shift (e.g., strong blur/out-of-focus, specular saturation, debris). In Case 1 (frame n) and Case 2 (frame n), the target is present in the ground truth but both networks predict all background. Importantly, the probability maps saturate

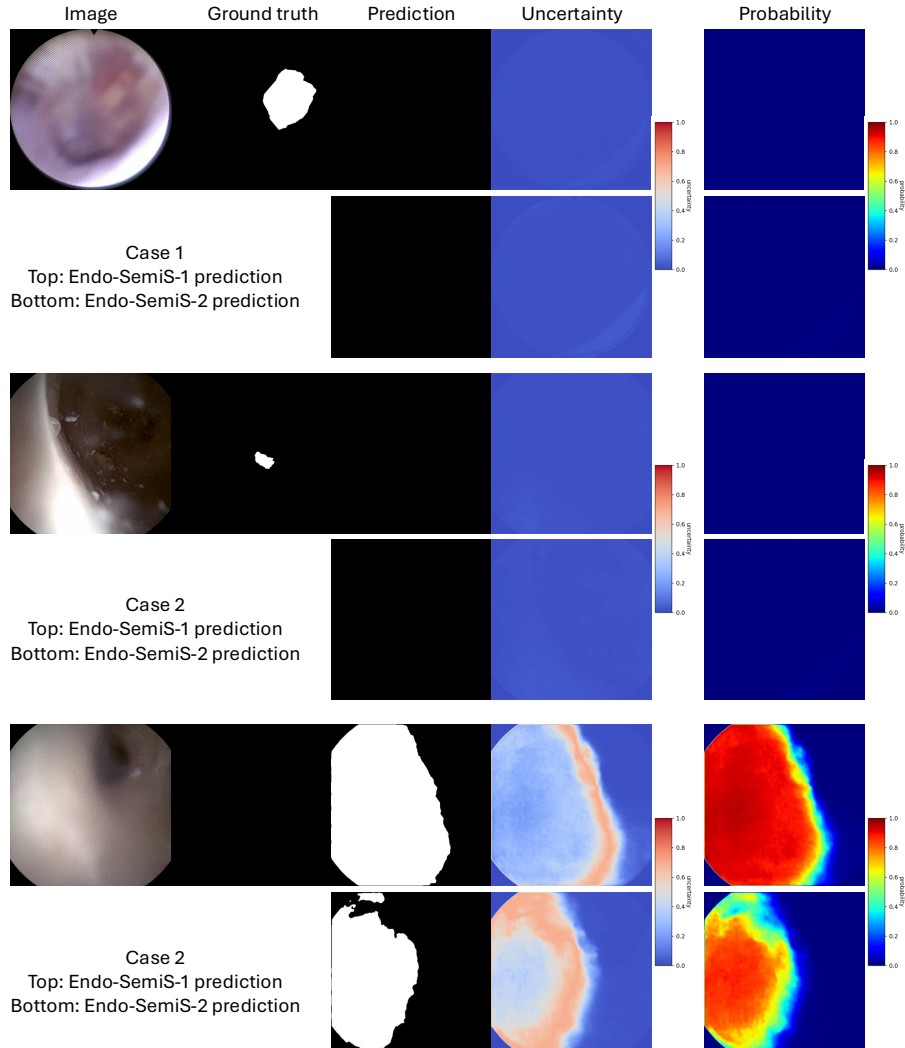

Figure 6: Failure cases. Each row shows an endoscopic frame, the ground-truth mask, the predicted binary image, the uncertainty map, and the probability map. For each section, the top and bottom panels correspond to Endo-SemiS-1 and Endo-SemiS-2, respectively. Color bars are normalized to $[0, 1]$ (blue: low, red: high). Endo-SemiS produces confidently wrong prediction.

toward background and the uncertainty (predictive entropy) remains near zero across the missed target region, with only weak increases near the predicted field-of-view boundary. Because our filtering rule relies on elevated uncertainty to reject unreliable pseudo-label pixels, these confidently wrong predictions are not filtered. Moreover, when both networks make the same confident error, the co-training signal provides no corrective disagreement, so cross-supervision cannot self-correct. Case 2 (frame m) further shows that uncertainty concentrates mainly at decision boundaries (higher entropy along edges) while remaining

low in the interior, which limits its ability to flag globally unreliable predictions when the visual pattern is far from the training distribution.

## Appendix E. Qualitative results

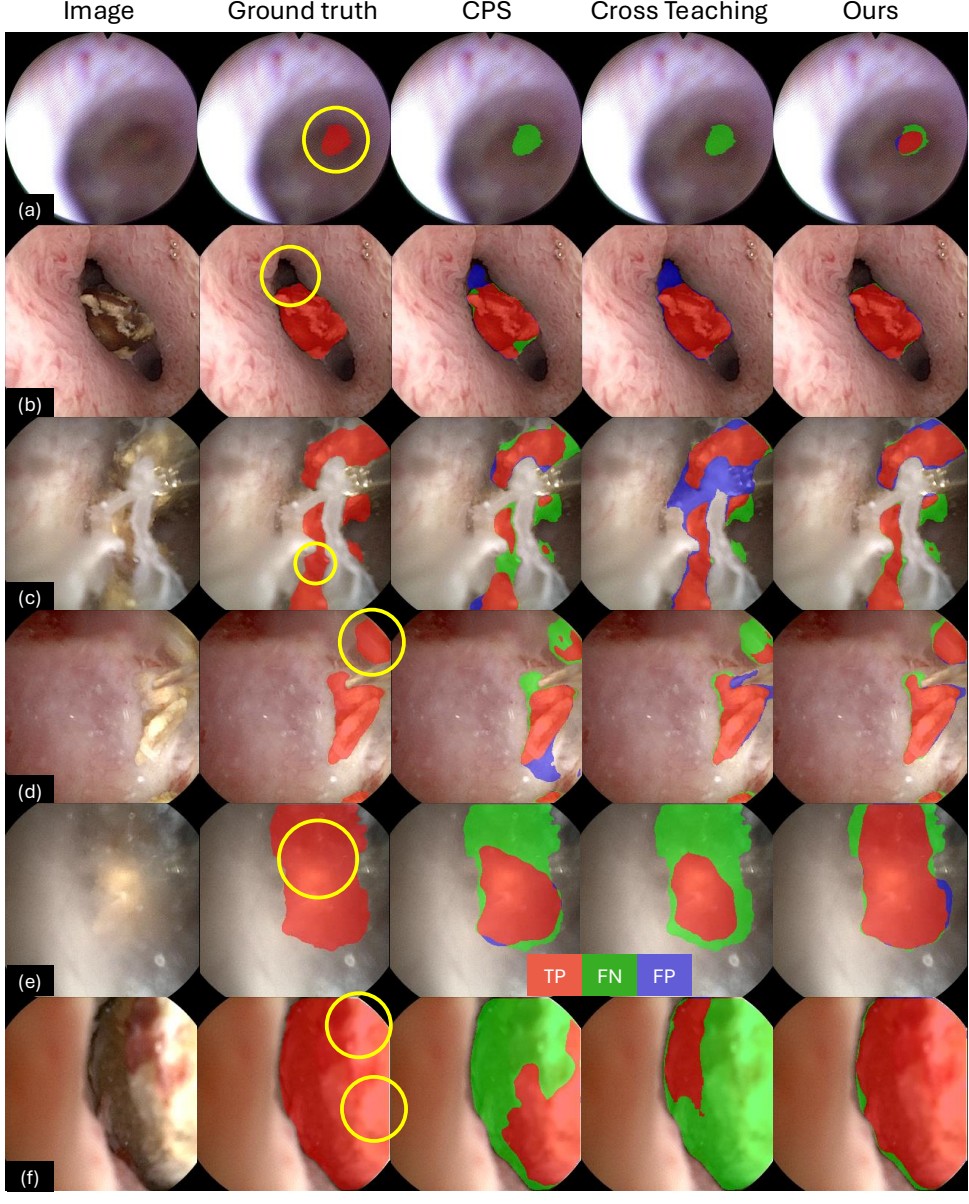

Figure 7: Qualitative kidney stone results (10% labeled data), larger version of Fig. 4 for better visualization. Yellow circles highlight poor visibility areas. (a) fiberoptic frames, (b) digital frames, (c) fluid distortions, (d) motion blur, (e) debris during stone ablation, and (f) illumination changes.

