# OpenReview forum: "Endo-SemiS: Towards Robust Semi-supervised Image Segmentation for Endoscopic Video"
_MIDL.io/2026/Conference — MIDL 2026 Poster_

### Official Review · Reviewer_1j2z · 2026-01-06

**Confidence:** 4
**Preliminary Rating:** 5

**Summary:**

The paper proposes a semi-supervised segmentation framework tailored for endoscopic video with limited annotations. The method combines four complementary ideas: (1) dual-network cross-supervision with independent U-Nets; (2) uncertainty-guided pseudo-labelling that fuses aleatoric (weak–strong augmentation) and epistemic (MC dropout) uncertainty with a dynamic, per-image threshold; (3) joint pseudo-label supervision that fuses predictions from both networks using pixel-wise uncertainty selection and then filters them; and (4) multi-level mutual learning aligning encoders and decoders on labeled data to reduce variance and stabilise pseudo-label quality. A plug-and-play spatiotemporal correction module further reduces frame-level FP/FN inconsistencies by leveraging short temporal windows at inference. The approach is validated on two clinical applications with consistent and substantial improvements over strong semi-supervised and fully supervised baselines, even surpassing full-supervision in some settings.

**Strengths:**

1. Novel integration of dynamic uncertainty filtering and joint pseudo-label fusion that directly targets noisy endoscopic frames.

2. Multi-level mutual learning aligns representations and logits, while the architecture design is well-justified for stabilising pseudo-label generation.

3. Strong, comprehensive empirical results across label ratios, with consistent inter-model performance and useful spatiotemporal post-processing.

4. Clear methodological exposition and figures; code availability and reasonable training details support reproducibility.

**Weaknesses:**

1. MC dropout passes add overhead; therefore, a discussion on latency for real-time deployment would be valuable.

2. While dual-network diversity is beneficial, an analysis of architectural heterogeneity (e.g., lightweight CNN vs. CNN variants) might enhance robustness further.

3. Sensitivity analysis on the dynamic thresholding rule and failure cases (e.g., extremely confident but wrong predictions) could strengthen robustness.

**Detailed Comments:**

1. The paper is well-written and easy to follow, with clear figures that map the contribution components (cross-supervision baseline, uncertainty-guided filtering, joint fusion, mutual learning).

2. The motivation for each component is explicitly tied to known SSL failure modes (confirmation bias, noisy pseudo-labels, variance between models), making the narrative cohesive.

3. Minor improvements can be made regarding the dynamic thresholding rule: lacking sensitivity analysis; more discussion on failure modes for confident-but-wrong predictions would help understand edge cases.

**Justification Of The Preliminary Rating:**

This submission makes a well-motivated, technically cogent, and practically impactful contribution to semi-supervised endoscopic segmentation. The integration of dynamic uncertainty filtering, joint pseudo-label fusion, and multi-level mutual learning is novel and effective, and the additional spatiotemporal correction addresses real deployment artefacts. Strong empirical results across two clinically important tasks, especially in low-label regimes, justify strong acceptance.

**Questions To Address In The Rebuttal:**

How sensitive are results to K (number of MC dropout passes) and to the specific locations of dropout layers in the decoder?

---

> ### Author Response · Authors · 2026-01-25
>
> We thank the reviewer for the constructive comments.
>
> **Threshold analysis (added Appendix D).** Table 6 investigates the sensitivity of Endo-SemiS to the uncertainty-mask thresholding rule on PolypGen (10% labeled). The two fixed thresholds capture complementary behaviors, μ+σ and P95. Our method combines these two fixed choices via a dynamic threshold T = min(μ(U) + σ(U), P95(U)) which automatically selects the stricter criterion per image. The combined rule achieves the best overall results in Table 6, including the top single-frame Dice for EndoSemi-S-2 (79.4) and strong sequence performance.
>
> **MC-dropout forward passes (added Table 6).** Increasing K from 5 to 10 can further improve sequence Dice but not frame Dice. Larger K does not provide consistent additional improvements, suggesting that moderate K is sufficient.
>
> |  |  | Single frame data |  | Sequence frame data |  |
> |---|---|---|---|---|---|
> | Threshold | K | EndoSemiS-1 | EndoSemiS-2 | EndoSemiS-1 | EndoSemiS-2 |
> | μ+σ | 5 | 76.1 | 79.1 | 69.8 | 68.9 |
> | P95 | 5 | 78.3 | 77.4 | 69.6 | 70.1 |
> | min(μ+σ, P95) | 5 | 76.2 | 79.4 | 69.3 | 71.2 |
> | min(μ+σ, P95) | 10 | 77.3 | 75.0 | 72.7 | 70.6 |
> | min(μ+σ, P95) | 15 | 76.0 | 74.8 | 70.5 | 69.8 |
>
> **Real-time deployment and inference dropout.** We thank the reviewer for this important practical consideration. We clarify that MC dropout is used only during training to estimate uncertainty for pseudo-label weighting, it is not used during inference. At test time, our method performs a single forward pass identical to standard supervised model without adding additional latency overhead.
>
> We benchmarked inference speed on an NVIDIA RTX A6000:
>
> | Resolution | Deployment FPS | Latency |
> |---|---|---|
> | 256×256 | 135.5 | 7.4 ms |
> | 512×512 | 72.6 | 13.8 ms |
>
> Both exceed the 25-30 FPS requirement for real-time endoscopy.
>
> For training, we used 8 dropout layers after each convolutional layer. Due to the limit time during rebuttal, we are unable to complete the ablation study on their location, and we'll leave this as future work.
>
> **Architectural heterogeneity (Table 5).** We thank the reviewer for this comment. From Table 5, we found:
>
> 1. Our framework is model-agnostic: it does not replace the backbone’s segmentation ability, but builds on it. With a stronger backbone (DSHNet), we preserve its strong supervised performance and further improve it through semi-supervised learning. With a weaker backbone (U-Net), we still provide consistent gains over supervised training, showing that the benefit is not tied to a specific architecture.
>
> 2. Our framework addresses the overfitting issue that affects DSHNet network trained only on labeled frame data, as such models generalize poorly to sequence data. Our uncertainty-weighted pseudo-label learning regularizes the model using unlabeled data, preventing overfitting and improving sequence generalization regardless of backbone choice.
>
> 3. Regarding using different architectures within the same framework, the Cross Teaching employs heterogeneous networks, e.g., CNN + Swin transformer. However, as shown in our results, it produces inconsistent performance across tasks. It does not reliably outperform homogeneous setups. This suggests that architectural heterogeneity alone is not sufficient, and the learning framework design matters more than simply pairing different architectures. We will further explore this setting in our future work.
>
> **Lightweight CNN (added Table 5).** We conducted an experiment with lightweight CNN, EfficientNet. Here are our key observations:
> - Baseline instability: The standalone EfficientNet baseline shows undesired results (e.g., for 10% labeled data, it produces 68±33 Dice for single-frame, 44±39 for sequence data), suggesting training instability or convergence issues with limited labeled data.
> - Endo-SemiS stabilizes EfficientNet: EfficientNet becomes trainable and achieves reasonable performance: Endo-SemiS-1 (EfficientNet): 73±31 (single) to 70±36 (sequence); Endo-SemiS-2 (EfficientNet): 74±32 (single) to 71±36 (sequence).
> - Smaller single-to-sequence gap: EfficientNet with Endo-SemiS shows only a ~3-point drop from single-frame to sequence evaluation, compared to DSHNet's ~5-7 point drop, suggesting potentially more consistent predictions despite lower absolute performance.
>
> **Failure case (added Appendix D).** We acknowledge a limitation of our uncertainty-guided pseudo-labeling: under severe appearance shift, both networks can produce the same confident but incorrect prediction, including missing true targets (false negatives) or hallucinating targets (false positives), while predictive entropy remains near-zero, which can evade our entropy-based filtering and offers no disagreement signal for cross-supervision to correct. The details can be viewed in Appendix D.

---

### Official Review · Reviewer_wpTV · 2026-01-06

**Confidence:** 5
**Preliminary Rating:** 2
**Final Rating:** 4

**Summary:**

This paper proposes a semi-supervised endoscopic video segmentation framework that combines dual-network cross-supervision, uncertainty-guided pseudo-labeling, joint pseudo-label supervision, and multi-level mutual learning to better exploit abundant unlabeled frames. Experiments on kidney stone ureteroscopy (in-house dataset) and cross-center polyp colonoscopy segmentation, reporting consistent gains over multiple SSL baselines with limited labels (e.g., 10%).

**Strengths:**

1.	Clear motivation. The paper clearly articulates why standard SSL can fail under high uncertainty and domain shift in endoscopic settings, then maps each limitation to a specific design choice (cross-supervision + uncertainty + joint labels + mutual learning).

2.	Results compare against many recognized SSL baselines (e.g., FixMatch, UniMatch, Mean Teacher, CPS, Cross Teaching) with clear reporting of metrics. On the kidney dataset with 10% labels, Endo-SemiS reports higher Dice than competing SSL methods and even exceeds the reported “upper bound” supervised U-Net in that table (87.6 vs. 85.3 Dice).

**Weaknesses:**

1 In Table 1, why is the Dice score of nnU-Net worse than that of U-Net?

2 Only U-Net and nnU-Net are included as baselines; more supervised baselines are needed for comparison.

3  In Table 4, no SSL baselines are included, and only U-Net results are reported.

4  Only two datasets are used to evaluate the proposed method: a kidney stone ureteroscopy in-house dataset and a cross-center polyp colonoscopy segmentation dataset. More public datasets are needed to further validate the method (e.g., SUN-SEG).

**Detailed Comments:**

See weakness.

**Justification Of Final Rating:**

The authors have addressed most of my concerns. Besides, the other two reviewers also recommend to accept (the positive comments from other reviewers). Therefore, I would like change my final rating from 2 to 4.

**Justification Of The Preliminary Rating:**

This paper proposes a semi-supervised endoscopic video segmentation framework that combines dual-network cross-supervision, uncertainty-guided pseudo-labeling, joint pseudo-label supervision, and multi-level mutual learning to better exploit abundant unlabeled frames. Experiments on kidney stone ureteroscopy (in-house dataset) and cross-center polyp colonoscopy segmentation, reporting consistent gains over multiple SSL baselines with limited labels (e.g., 10%). However, there are still some concerns listed in weakness. Therefore, I suggest weak reject.

**Questions To Address In The Rebuttal:**

See weakness.

---

> ### Author Response · Authors · 2026-01-25
>
> We thank the reviewer for comments.
>
> **Performance between nnUNet and UNet.** Thanks for raising this question.
>
> The performance difference is small (80.5% vs 79.5%) and within the standard deviation, suggesting it is not statistically significant. We attribute this marginal gap to the limited labeled data: nnU-Net default setting may lead to slight overfitting when supervision is scarce. This is supported by the upper bound results, where nnU-Net (85.5%) performs comparably to U-Net (85.3%). We intentionally used nnU-Net with its default self-configuring settings without additional tuning, as its main strength is providing a strong out-of-the-box baseline.
>
> In addition, our U-Net implementation achieves 79% and 69% Dice on PolypGen single-frame and sequence test sets respectively (Table 5), matching the official benchmark [1] on single frames (79%) and exceeding it on sequences (66%).
>
> [1] A multi-centre polyp detection and segmentation dataset for generalisability assessment
>
> **Supervised and SSL baselines (added Table 5).** Following the reviewer's feedback, we have added supervised and SSL baselines to provide a more comprehensive comparison. The updated results can be found in the new Table 5 and the generalizability analysis on page 11.
>
> We've added nnUNet and additional strong supervised baselines for comparison. We also include PNS+ [1], the top-performing method on the SUN-SEG dataset, and DSHNet [2], a recent state-of-the-art polyp segmentation model.
>
> [1] Video polyp segmentation: A deep learning perspective \
> [2] Dynamic spectrum-driven hierarchical learning network for polyp segmentation
>
> Among supervised baselines with 10% labels, DSHNet achieves the best single-frame Dice (80%), while U-Net performs best on sequences (64%). The nnUNet results are lower than U-Net, which may reflect overfitting as explained in W1. Also DSHNet may be overfitted since it produces low Dice on sequence data.
>
> For semi-supervised methods, we added CPS and Cross Teaching to Table 5 for comparison.  Endo-SemiS with U-Net backbone achieves 79% and 71% Dice on single-frame and sequence test sets. When combined with DSHNet backbone, Endo-SemiS achieves the best sequence performance (73% Dice), outperforming all baselines including the fully supervised upper bounds.
>
> In addition, we ran a lightweight CNN (EfficientNet) shows limited performance under supervised training with limited labels. Yet when trained with Endo-SemiS, it achieves competitive sequence performance comparable to heavier backbones while exhibiting the smallest gap between single-frame and sequence evaluation.
>
> |  | Single frame data |  |  | Sequence frame data |  |  |
> |---|---|---|---|---|---|---|
> | Methods | Dice | Sen. | Spe. | Dice | Sen. | Spe. |
> | U-Net | 75±30 | 73±31 | 100±1 | 64±38 | 64±38 | 100±1 |
> | nnUNet | 75±33 | 78±32 | 99±3 | 53±43 | 68±40 | 99±4 |
> | PNS+ | 71±34 | 71±35 | 99±2 | 48±41 | 47±42 | 99±3 |
> | DSHNet | 80±24 | 87±24 | 99±1 | 59±39 | 76±31 | 99±2 |
> | EfficientNet | 67±33 | 69±35 | 99±2 | 44±39 | 41±37 | 99±2 |
> | CPS-1 | 77±32 | 74±33 | 100±1 | 68±40 | 65±40 | 100±1 |
> | CPS-2 | 75±34 | 71±34 | 100±1 | 64±42 | 65±42 | 100±1 |
> | Cross Teaching-1 | 73±35 | 70±36 | 100±1 | 63±41 | 60±41 | 100±1 |
> | Cross Teaching-2 | 75±34 | 73±34 | 100±1 | 65±40 | 63±40 | 100±1 |
> | Endo-SemiS-1 | 76±34 | 75±34 | 100±1 | 69±39 | 67±39 | 100±1 |
> | Endo-SemiS-2 | 79±30 | 77±31 | 100±1 | 71±37 | 70±37 | 100±2 |
> | Endo-SemiS-1 (DSHNet) | 80±28 | 82±29 | 100±1 | **73±34** | 73±34 | 99±1 |
> | Endo-SemiS-2 (DSHNet) | 78±30 | 82±29 | 99±1 | **73±35** | 73±34 | 100±1 |
> | Endo-SemiS-1 (EfficientNet) | 73±31 | 81±32 | 99±2 | 70±36 | 73±35 | 99±2 |
> | Endo-SemiS-2 (EfficientNet) | 74±32 | 79±33 | 99±2 | 71±36 | 71±36 | 99±2 |
> | Upper bound U-Net | 79±30 | 79±31 | 99±2 | 69±37 | 74±35 | 99±2 |
> | Upper bound nnUNet | 80±29 | 84±26 | 99±1 | 62±41 | 71±38 | 99±2 |
> | Upper bound PNS+ | 74±32 | 77±33 | 99±3 | 49±42 | 49±43 | 99±3 |
> | Upper bound DSHNet | **85±24** | 92±19 | 99±2 | 66±39 | 79±33 | 99±3 |
> | Upper bound EfficientNet | 75±32 | 77±34 | 99±1 | 45±40 | 41±40 | 100±1 |
>
> **SUN-SEG dataset.** Due to the limited time window for the rebuttal, we are not able to include a comprehensive analysis on SUN-SEG. However, to alleviate this concern within the available time, we instead include the top-performing model on SUN-SEG (PNS+), a strongly supervised model, and evaluate it under same protocol on PolypGen (see our response above on supervised baselines). We will conduct experiments on SUN-SEG in the future work.
>
> | Model | Train Data | Frame Test (Dice) | Seq Test (Dice) |
> |---|---|---|---|
> | **Sequence Data Training** |  |  |  |
> | Zero-shot (pre-trained) | SUN-SEG | -- | 52.6 ± 40.8 |
> | Scratch | seq1-15 | -- | 46.9 ± 44.9 |
> | Fine-tune | seq1-15 | -- | 59.7 ± 43.6 |
> | **Single Frame Training** |  |  |  |
> | Scratch | c1-c5 | 74.5 ± 34.7 | 46.7 ± 43.5* |
> | Fine-tune | c1-c5 | 74.0 ± 32.2 | 49.4 ± 41.9* |
> \* denotes that the repeated frames are used for inference.

---

### Official Review · Reviewer_VAmw · 2026-01-09

**Confidence:** 3
**Preliminary Rating:** 4
**Final Rating:** 5

**Summary:**

This manuscript proposes a framework for improving image segmentation for endoscopic video using semi-supervised technologies. The framework has multiple strategies, including cross-supervision, pseudo-label generation and aggregation, and mutual learning. The results show the best or second-best results in almost all settings.

**Strengths:**

+ The topic of this manuscript is timely and interesting.
+ The proposed method is sound. The design of loss functions is convincing.
+ The experiment includes public and in-house datasets. Multiple competing methods are included.
+ Codes are released.

**Weaknesses:**

- It is understandable that the authors intentionally reduced the proportion of labeled data in the experimental design to highlight the efficacy of semi-supervised learning and pseudo-labels. However, it is better to have further discussion for the clinical practicality of this setting. For example, the authors could briefly comment on the performance when the labeled data ratio is 50% or 70%. Detailed quantitative results are not strictly necessary, but a textual discussion would be sufficient. In addition, similar to Figures 4 and 5, providing several visualization examples of failure cases would offer a more deployment-oriented and clinically friendly perspective.

- A dedicated Limitations section would be welcome.

- If there were a table showing the segmentation performance in different region sizes (small, medium, large regions in terms of ground truth pixels), it would be much better.

**Detailed Comments:**

- In the caption of Table 1, the authors may consider adding a note directing readers to Appendix A for further details about the competing methods.

- The authors may consider changing the orders of methods in Figures 4 and 5, putting Ours last.

**Justification Of Final Rating:**

Thanks for the authors' response. The authors have addressed all my concerns, including major concerns and minor concerns. I have changed my rating from Weak accept to Strong accept. This paper has already been in great shape.

**Justification Of The Preliminary Rating:**

The semi-supervised method of this manuscript is sound, showing the authors' strong experience and insights. The results are convincing, even though there are some concerns. This manuscript is easy to follow.

**Questions To Address In The Rebuttal:**

The authors can address all concerns in Weaknesses and Detailed Comments.

---

> ### Author Response · Authors · 2026-01-25
>
> We thank the reviewer for the comments.
>
> **Label ratio.** We thank the reviewer's suggestion to discuss higher labeled ratios (e.g., 50%/70%). Our experiments emphasize a low-label strategy because dense, frame-wise annotation in endoscopic videos is costly and often infeasible in practice, since some videos are hours long with a sample rate of 30 FPS. Clinically, labels are more commonly available at sparse keyframes or limited cases paired with unlabeled video.
>
> As the labeled ratio increases, performance typically approaches saturation and the relative improvement of semi-supervised learning over strongly supervised baselines tends to diminish. Nevertheless, our framework remains applicable as a plug-and-play training strategy even when more labels are available; in this scenario its primary benefit shifts from maximizing mean Dice to improving robustness against noisy supervision and domain shifts that frequently arise in endoscopic imaging.
>
> Concretely, this benefit is reflected by our pseudo-label correction mechanism: Fig. 3 illustrates how uncertainty-aware filtering and cross-model agreement suppress noisy regions in raw pseudo-labels, while Fig. 4 visualizes the subsequent refinement/aggregation that yields cleaner supervision signals. These figures highlight that our framework is designed to improve pseudo-label correctness. This behavior remains useful even when more labeled data is available, since endoscopic videos still exhibit substantial noise (e.g., specularity, motion blur) and domain shifts.
>
> **Failure case (added Appendix D).** We acknowledge a limitation of our uncertainty-guided pseudo-labeling: under severe appearance shift, both networks can produce the same confident but incorrect prediction, including missing true targets (false negatives) or hallucinating targets (false positives), while predictive entropy remains near-zero. This can evade our entropy-based filtering and offers no disagreement signal for cross-supervision to correct. We also mention this in the conclusion section as a limitation.
>
> **Kidney stone size analysis (added Table 4).** We stratified the test set (n=3,959) by kidney stone size based on ground truth mask area relative to image dimensions H × W: No Stone (n=1,364), Small (< ⅛ H×W, n=495), Medium (⅛–¼ H×W, n=418), and Large (≥ ¼ H×W, n=1,682). As shown in Table 4, Endo-SemiS achieves the best performance on small, medium, and large groups. Among SSL methods, the largest improvement is for large stones. In intra-operative settings, close-range views are common and often involve ongoing ablation, which increases boundary ambiguity and makes segmentation more challenging. Compared to cross-supervised methods, the larger improvements on small stones suggest better robustness on challenging small-region segmentation where limited pixel support makes predictions sensitive to noise. Overall, these results show that the proposed uncertainty-guided learning remains effective across stone sizes.
>
> | Method | Label | No Stone (n=1364) | Small (n=495) | Medium (n=418) | Large (n=1682) | Overall |
> |--------|-------|-------------------|---------------|----------------|----------------|---------|
> | U-Net | 100% | 91.3±29.6 | 63.8±38.5 | 82.0±25.9 | 87.6±22.8 | 85.3±29.2 |
> | nnU-Net | 100% | 90.2±29.1 | 64.9±38.9 | 82.2±25.3 | 88.6±21.6 | 85.5±28.5 |
> | U-Net | 10% | 75.4±42.5 | 65.6±35.2 | 82.9±20.5 | 88.4±16.8 | 80.5±32.1 |
> | nnU-Net | 10% | 80.9±39.4 | 57.6±40.6 | 77.4±29.6 | 85.3±23.7 | 79.5±33.8 |
> | UniMatch | 10% | 89.0±31.3 | 69.2±35.7 | 81.4±25.0 | 88.5±19.3 | 85.5±27.6 |
> | CPS | 10% | 88.3±32.2 | 65.7±37.0 | 83.1±22.0 | 88.9±18.6 | 85.2±28.0 |
> | Cross Teaching | 10% | **93.5±24.6** | 65.9±38.6 | 83.8±26.0 | 87.1±24.3 | 85.6±28.7 |
> | **Endo-SemiS (Ours)** | 10% | 90.5±29.4 | **70.4±36.0** | **84.4±23.0** | **91.1±17.9** | **87.6±26.4** |
>
> *Table 4: Dice score (%) stratified by kidney stone size. Size categories are defined by ground truth mask area relative to the image area (H×W). Best semi-supervised results are in bold.*
>
> **Detailed comment.** We thank the reviewer for these suggestions and made the corresponding changes. We'll change the figure order and update the revised manuscript.

---

### Author Rebuttal · Authors · 2026-01-25

**Rebuttal:**

We thank the reviewers for their detailed and constructive comments. We have addressed them point-by-point and uploaded the revised manuscript.

**Supporting Material:**

/attachment/ee43d5749ab208a5920ddc8451c0b1c51197656e.pdf

---

### Meta-Review · Area_Chair_gyAx · 2026-02-07

**Recommendation:** Accept (Oral)
**Confidence:** 4

**Metareview:**

This paper presents a well-motivated semi-supervised framework for endoscopic video segmentation, combining cross-supervision, uncertainty-guided pseudo-labeling, joint fusion, and mutual learning to effectively address key challenges in low-label settings. The method is clearly described, extensively evaluated against strong SSL baselines, and consistently achieves state-of-the-art performance. All reviewer concerns were carefully addressed in the rebuttal.

---

### Decision · Program_Chairs · 2026-02-13

Accept (Poster)